# Fundamentals of Physics-Informed Neural Networks Applied to Solve the Reynolds Boundary Value Problem

**Andreas Almqvist**

Division of Machine Elements, Luleå University of Technology, SE-971 87 Luleå, Sweden; andreas.almqvist@ltu.se

**Abstract:** This paper presents a complete derivation and design of a physics-informed neural network (PINN) applicable to solve initial and boundary value problems described by linear ordinary differential equations. The objective with this technical note is not to develop a numerical solution procedure which is more accurate and efficient than standard finite element- or finite difference-based methods, but to give a fully explicit mathematical description of a PINN and to present an application example in the context of hydrodynamic lubrication. It is, however, worth noticing that the PINN developed herein, contrary to FEM and FDM, is a meshless method and that training does not require big data which is typical in machine learning.

**Keywords:** PINN; machine learning; reynolds equation

## 1. Introduction

There are various categories of artificial neural networks (ANN) and a physics-informed neural network (PINN), see [1] for a recent review on the matter, is a neural network trained to solve both supervised and unsupervised learning tasks while satisfying some given laws of physics, which may be described in terms of nonlinear partial differential equations (PDE). For example, the balance of momentum and conservation laws in solid- and fluid mechanics and various types of initial value problems (IVP) and boundary value problems (BVP), see e.g., [2,3]. The application of a PINN (of this type) to solve differential equations, renders meshless numerical solution procedures [4], and an important feature from a machine learning perspective, is that it is not a data-driven approach requiring a large set of training data to learn the solution.

In fluid mechanics, under certain assumptions, i.e., that the fluid is incompressible, iso-viscous, the balance of linear momentum and the continuity equation, for flows in narrow interfaces reduces to the classical Reynolds equation [5]. For more recent work establishing lower-dimensional models in a similar manner, see e.g., [6–8]. The present work describes how a PINN can be adapted and trained to solve both initial and boundary value problems, described by ordinary differential equations, numerically. The theoretical description starts by presenting the neural network's architecture and it is first applied to solve an initial value problem, which is described by a first order ODE, which can be solved analytically so that the validity of the solution can be thoroughly assessed. Thereafter, it is used to obtain a PINN for the classical one-dimensional Reynolds equation, which is a boundary value problem governing, e.g., the flow of lubricant between the runner and the stator in a 1D slider bearing. The novelty and originality of the present work lays in the explicit mathematical description of the cost function, which constitutes the *physics-informed* feature of the ANN, and the associated gradient with respect to the networks weights and bias. Important features of this particular numerical solution procedure, that is publicly available here: [9], are that it is not data driven, i.e., no training data need to be provided and that it is a meshless method [4].

## 2. PINN Architecture

Knowing the characteristics of the solution to the differential equation under consideration is very helpful when designing the PINN architecture, including structure, number of hidden layers, activation function, etc. For this reason, the PINN developed here has one input node $x$ (the independent variable representing the spatial coordinate), one hidden layer consisting of $N$ nodes and one output node $y$ (the dependent variable representing pressure). Figure 1 depicts a graphical illustration of the present architecture, which when trained solves both the IVP example and the Reynolds BVP considered here.

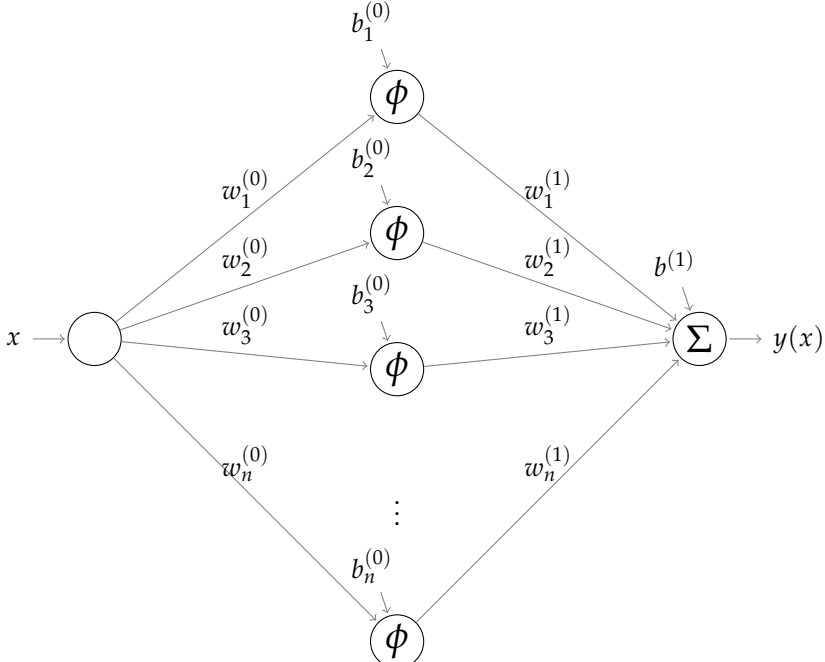

**Figure 1.** Architecture of the PINN employed to solve the IVP and BVP considered here.

The Sigmoid function, i.e.,

$$\phi(\xi) = \frac{1}{1 + e^{-\xi}}, \tag{1}$$

which is mapping $\mathbb{R}$ to $[0, 1]$ and exhibits the property

$$\phi'(\xi) = \phi(\xi)(1 - \phi(\xi)). \tag{2}$$

is employed as activation function for the hidden layer. This means that the neural network has $3N + 1$ trainable parameters. That is, the weights $w_i^{(0)}$ and bias $b_i^{(0)}$ for the nodes in the hidden layer and the weights $w_i^{(1)}$, $i = 1 \dots N$, for each synapses connecting them with the output node, plus the bias $b^{(1)}$ applied there.

Based on this particular architecture, the output $z_i$ of each node in the first hidden layer is,

$$z_i(x) = \phi\left(w_i^{(0)}x + b_i^{(0)}\right). \tag{3}$$

The output value is then given by applying the Sigmoid activation function scaled by the weight from the node in the second layer and yields

$$y(x) = b^{(1)} + \sum_{i=1}^{N} w_i^{(1)} z_i(x) = b^{(1)} + \sum_{i=1}^{N} w_i^{(1)} \phi\left(w_i^{(0)}x + b_i^{(0)}\right). \tag{4}$$

Let us now construct the cost function which the network will be trained to minimise. While the cost function appearing in a typical machine learning procedure is just the

quadratic difference between the predicted and the target values, it will here be defined by means of the operators $\mathcal{L}$ and $\mathcal{B}$. The cost function applied here reads

$$l = \left\langle (\mathcal{L}y - f)^2 \right\rangle + ((\mathcal{B}y - \mathbf{b}) \cdot \mathbf{e_1})^2 + ((\mathcal{B}y - \mathbf{b}) \cdot \mathbf{e_2})^2, \tag{5}$$

where $\langle f \rangle$ defines the average value of $f$, and this is exactly the feature that makes an ANN "physics informed", i.e., a PINN.

Since $\mathcal{L}y$ is a differential operator the cost function contains derivatives of the network output (4). In order to obtain an expression of the cost function, in terms of the input $x$, the weights $w$ and bias $b$, the network output (4), must be differentiated twice with respect to (w.r.t. ) $x$. This can be accomplished by some kind of automatic differentiation (AD) (also referred to as algorithmic differentiation, computer differentiation, auto-differentiation or simply autodiff), which is a computerised methodology based on the chain rule, which can be applied to efficiently and accurately evaluate derivatives of numeric functions, see e.g., [10,11]. The present work instead applies symbolic differentiation to clearly explain all the essential details of the PINN. Indeed, differentiating one yield

$$y'(x) = \frac{\partial}{\partial x} \left( \left( \sum_{i=1}^{N} w_i^{(1)} z_i(x) \right) + b^{(1)} \right) = \frac{\partial}{\partial x} \left( \left( \sum_{i=1}^{N} w_i^{(1)} \phi \left( w_i^{(0)} x + b_i^{(0)} \right) \right) + b^{(1)} \right) =$$
$$= \sum_{i=1}^{N} w_i^{(1)} w_i^{(0)} \phi' \left( w_i^{(0)} x + b_i^{(0)} \right) = \sum_{i=1}^{N} w_i^{(1)} w_i^{(0)} \phi \left( w_i^{(0)} x + b_i^{(0)} \right) \left( 1 - \phi \left( w_i^{(0)} x + b_i^{(0)} \right) \right), \tag{6}$$

and, because of (2), a consecutive differentiation then yields

$$y''(x) = \frac{\partial}{\partial x} \left( \sum_{i=1}^{N} w_i^{(1)} w_i^{(0)} \phi' \left( w_i^{(0)} x + b_i^{(0)} \right) \right) = \sum_{i=1}^{N} w_i^{(1)} \left( w_i^{(0)} \right)^2 \phi'' \left( w_i^{(0)} x + b_i^{(0)} \right) =$$
$$= \sum_{i=1}^{N} w_i^{(1)} \left( w_i^{(0)} \right)^2 \phi' \left( w_i^{(0)} x + b_i^{(0)} \right) \left( 1 - 2\phi \left( w_i^{(0)} x + b_i^{(0)} \right) \right) = \tag{7}$$
$$= \sum_{i=1}^{N} w_i^{(1)} \left( w_i^{(0)} \right)^2 \phi \left( w_i^{(0)} x + b_i^{(0)} \right) \left( 1 - \phi \left( w_i^{(0)} x + b_i^{(0)} \right) \right) \left( 1 - 2\phi \left( w_i^{(0)} x + b_i^{(0)} \right) \right).$$

Moreover, finding the set of weights and bias minimising the cost function requires its partial derivatives w.r.t. to each weight and bias defining the PINN. In the subsections below, we will present how to achieve this, by first considering a first order differential equation with an analytical solution, and, thereafter, we will consider the classical Reynolds equation which is a second order (linear) ODE that describes laminar flow of incompressible and iso-viscous fluids in narrow interfaces.

### 3. A First Order ODE Example

Let us consider the first order ODE, describing the initial value problem (IVP) given by

$$\mathcal{L}y - f = y' + 2xy = 0, \quad x > 0 \tag{8a}$$
$$\mathcal{B}y - \mathbf{b} = y(0) - 1 = 0, \tag{8b}$$

with the exact solution $y = e^{-x^2}$. By means of (6), a cost function suitable for solving (8) may be generated by

$$l = \left\langle \left[ \sum_{i=1}^{N} w_i^{(1)} w_i^{(0)} \phi \left( w_i^{(0)} x + b_i^{(0)} \right) \left( 1 - \phi \left( w_i^{(0)} x + b_i^{(0)} \right) \right) + \right. \right.$$
$$\left. \left. 2x \left( \left( \sum_{i=1}^{N} w_i^{(1)} \phi \left( w_i^{(0)} x + b_i^{(0)} \right) \right) + b^{(1)} \right) \right]^2 \right\rangle + [y(0) - 1]^2 \tag{9}$$

The solution of (8) can be obtained by implementing a training routine which iteratively finds the set of weights $w$ and bias $b$ that minimises (9) (and similarly for (19) minimising (17)). The most well-known of these is the Gradient Decent method attributed to Cauchy, who first suggested it in 1847 [12]. For an overview, see, e.g., [13].

As mentioned in the previous section, the derivatives of (4) w.r.t. to the weights $w$ and bias $b$ are required to find them, and *automatic differentiation* is, normally, employed to perform the differentiation. However, here we carry out symbolic differentiation to demonstrate exactly the explicit expressions that constitute the gradient of the cost function. Indeed, by taking the partial derivatives we obtain

$$\frac{\partial y}{\partial w_i^{(0)}} = \frac{\partial}{\partial w_i^{(0)}} \left( \left( \sum_{i=1}^{N} w_i^{(1)} \phi \left( w_i^{(0)} x + b_i^{(0)} \right) \right) + b^{(1)} \right) = w_i^{(1)} \phi' \left( w_i^{(0)} x + b_i^{(0)} \right) x, \quad (10a)$$

$$\frac{\partial y}{\partial w_i^{(1)}} = \frac{\partial}{\partial w_i^{(1)}} \left( \left( \sum_{i=1}^{N} w_i^{(1)} \phi \left( w_i^{(0)} x + b_i^{(0)} \right) \right) + b^{(1)} \right) = \phi \left( w_i^{(0)} x + b_i^{(0)} \right), \quad (10b)$$

$$\frac{\partial y}{\partial b_i^{(0)}} = \frac{\partial}{\partial b_i^{(0)}} \left( \left( \sum_{i=1}^{N} w_i^{(1)} \phi \left( w_i^{(0)} x + b_i^{(0)} \right) \right) + b^{(1)} \right) = w_i^{(1)} \phi' \left( w_i^{(0)} x + b_i^{(0)} \right), \quad (10c)$$

$$\frac{\partial y}{\partial b^{(1)}} = 1. \quad (10d)$$

Moreover, the derivatives of the cost function (5) w.r.t. to the weights and bias are also required. For the derivative w.r.t. $w_i^{(0)}$ for the first order ODE (8), this means that

$$\left\langle 2 \left( y' + 2xy \right) \left( \frac{\partial y'}{\partial w_i^{(0)}} + 2x \frac{\partial y}{\partial w_i^{(0)}} \right) \right\rangle + 2 (y(0) - 1) \frac{\partial y(0)}{\partial w_i^{(0)}}. \quad (11)$$

To complete the analysis, we also need expressions for the derivatives of $y'$ w.r.t. $w_i^{(0)}$, $w_i^{(1)}$, $b_i^{(0)}$ and $b^{(1)}$. By the chain rule, the following expressions can be obtained, viz.

$$\frac{\partial y'}{\partial w_i^{(0)}} = \frac{\partial}{\partial w_i^{(0)}} \sum_{i=1}^{N} w_i^{(1)} w_i^{(0)} \phi' \left( w_i^{(0)} x + b_i^{(0)} \right) =$$
$$= w_i^{(1)} \phi' \left( w_i^{(0)} x + b_i^{(0)} \right) + x w_i^{(1)} \left( w_i^{(0)} \right)^2 \phi'' \left( w_i^{(0)} x + b_i^{(0)} \right), \quad (12a)$$

$$\frac{\partial y'}{\partial w_i^{(1)}} = \frac{\partial}{\partial w_i^{(1)}} w_i^{(1)} w_i^{(0)} \phi' \left( w_i^{(0)} x + b_i^{(0)} \right) = w_i^{(0)} \phi' \left( w_i^{(0)} x + b_i^{(0)} \right), \quad (12b)$$

$$\frac{\partial y'}{\partial b_i^{(0)}} = \frac{\partial}{\partial b_i^{(0)}} \sum_{i=1}^{N} w_i^{(1)} w_i^{(0)} \phi' \left( w_i^{(0)} x + b_i^{(0)} \right) = w_i^{(1)} w_i^{(0)} \phi'' \left( w_i^{(0)} x + b_i^{(0)} \right), \quad (12c)$$

$$\frac{\partial y'}{\partial b^{(1)}} = 0. \quad (12d)$$

What remains now is to obtain expressions for $y(0)$ and the partial derivatives of $y(0)$, w.r.t. to the weights and bias. Let us start with $y(0)$. With $y(x)$ given by (4) we directly have

$$y(0) = \left( \sum_{i=1}^{N} w_i^{(1)} \phi \left( b_i^{(0)} \right) \right) + b^{(1)}, \quad (13)$$

which, in turn, means that

$$\frac{\partial y(0)}{\partial w_i^{(0)}} = 0, \tag{14a}$$

$$\frac{\partial y(0)}{\partial w_i^{(1)}} = \frac{\partial}{\partial w_i^{(1)}}\left(\left(\sum_{i=1}^{N} w_i^{(1)}\phi\left(b_i^{(0)}\right)\right) + b^{(1)}\right) = \phi\left(b_i^{(0)}\right), \tag{14b}$$

$$\frac{\partial y(0)}{\partial b_i^{(0)}} = \left(\left(\sum_{i=1}^{N} w_i^{(1)}\phi\left(b_i^{(0)}\right)\right) + b^{(1)}\right) = w_i^{(1)}\phi'\left(b_i^{(0)}\right) \tag{14c}$$

$$\frac{\partial y(0)}{\partial b^{(1)}} = 1. \tag{14d}$$

The PINN (following the architecture presented above) was implemented as computer program in MATLAB. The program was employed to obtain a numerical solution to the IVP in (8), using the parameters in Table 1.

**Table 1.** Parameters used to defined the PINN to for the IVP in (8).

| Parameter | Description | Value |
|-----------|-------------|-------|
| $N_i$ | # of grid points for the solution domain $[0, 2]$ | 41 |
| $N_e$ | # of training batches (# or corrections during 1 Epoch) | 1000 |
| $T_b$ | # of Epochs (1 Epoch contains $T_b$ training batches) | 100 |
| $L_r$ | Learning rate coefficient (relaxation for the update) | 0.01 |
| $N$ | # of nodes/neurons in the hidden layer | 10 |

The weights $w_i^{(0)}$ and bias $b_i^{(0)}$ were initialised using randomly generated and uniformly distributed numbers in the interval $[-2, 2]$, while the weights $w_i^{(1)}$ were initially set to zero and the bias $b^{(1)}$ to one, to ensure fulfilment of the initial condition ($y(0) = 1$).

Table 2 lists the weights an bias corresponding to the solution presented in Figure 2. We note that, with the weights and bias given by Table 2, the trained network's prediction exhibits the overall error

$$\frac{1}{N_i}\sqrt{\sum_{k=1}^{N_i}\left(e^{-x_k^2} - y(x_k)\right)^2} = 5.8 \times 10^{-4}, \tag{15}$$

and $1 - y(0) = 2.2 \times 10^{-4}$, when comparing against the initial condition.

**Table 2.** Parameters used to defined the PINN for the IVP (8).

| Node | $w^{(0)}$ | $b^{(0)}$ | $w^{(1)}$ | $b^{(1)}$ |
|------|-----------|-----------|-----------|-----------|
| 1 | 1.8500 | −0.5946 | −3.5805 | 0.3055 |
| 2 | 1.8588 | 1.5974 | 0.9712 | |
| 3 | 0.3025 | 1.9241 | 0.8921 | |
| 4 | 1.4546 | 0.3742 | −0.9955 | |
| 5 | 0.5065 | 1.2535 | −0.1430 | |
| 6 | −1.0898 | −1.0199 | −1.1067 | |
| 7 | −0.8302 | 0.3519 | −1.1668 | |
| 8 | 0.3789 | 1.6502 | 0.1754 | |
| 9 | 2.5012 | 0.7657 | 1.2955 | |
| 10 | 2.2743 | 1.4172 | 1.2787 | |

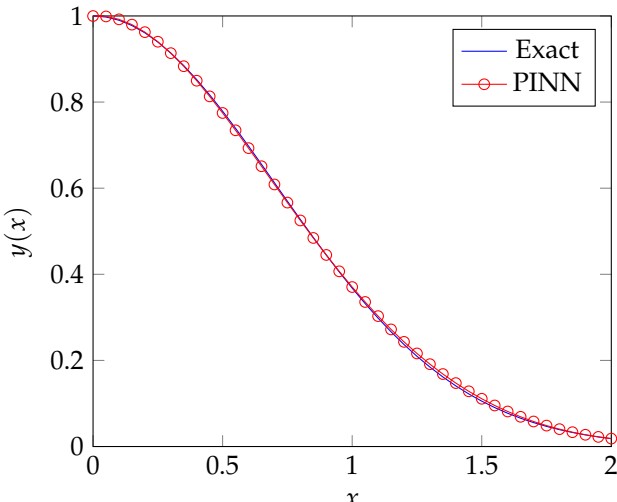

**Figure 2.** The solution to the IVP (8), predicted by the PINN (red line with circle markers) and the exact solution obtained by integration (blue continuous line).

### 4. A PINN for the Classical Reynolds Equation

The Reynolds equation for a one-dimensional flow situation, where the lubricant is assumed to be incompressible and iso-viscous, is a second order Boundary Value Problem (BVP), which in dimensionless form can be formulated as

$$\frac{d}{dx}\left(c(x)\frac{dy}{dx}\right) = f(x), \quad 0 < x < 1, \tag{16a}$$

$$y(0) = 0, \quad y(1) = 0, \tag{16b}$$

where $c(x) = H^3$, $f(x) = dH/dX$ and $H$ is the dimensionless film thickness, if it is assumed that the pressure $y$ at the boundaries is zero. For the subsequent analysis it is, however, more suitable work with a condensed form which can be obtained by defining the operators $\mathcal{L}$ and $\mathcal{B}$ as

$$\mathcal{L}y = c(x)y'' + c'(x)y', \tag{17a}$$

$$\mathcal{B}y = \begin{bmatrix} y(0) \\ y(1) \end{bmatrix}. \tag{17b}$$

The Reynolds BVP given by (16) can then be presented as

$$\mathcal{L}y - f = 0, \quad 0 < x < 1, \tag{18a}$$

$$\mathcal{B}y - \mathbf{b} = \mathbf{0}, \tag{18b}$$

where $\mathbf{b} = \mathbf{0}$.

For the Reynolds BVP, the cost function (5) becomes

$$l = \left\langle \left(c(x)y'' + c'(x)y' - f\right)^2 \right\rangle + y^2(0) + y^2(1), \tag{19}$$

and from the analysis presented for the IVP in Section 3 above, we have all the "ingredients" except for the partial derivatives of $y''$ and $y(1)$ w.r.t. to the weights and bias. For $y''$, based on (7) and (12), we obtain

$$\frac{\partial y''}{\partial w_i^{(0)}} = \frac{\partial}{\partial w_i^{(0)}} \sum_{i=1}^{N} w_i^{(1)} \left(w_i^{(0)}\right)^2 \phi'' \left(w_i^{(0)} x + b_i^{(0)}\right) =$$

$$= 2w_i^{(1)} w_i^{(0)} \phi'' \left(w_i^{(0)} x + b_i^{(0)}\right) + x w_i^{(1)} \left(w_i^{(0)}\right)^2 \phi''' \left(w_i^{(0)} x + b_i^{(0)}\right), \tag{20a}$$

$$\frac{\partial y''}{\partial w_i^{(1)}} = \frac{\partial}{\partial w_i^{(1)}} \sum_{i=1}^{N} w_i^{(1)} \left(w_i^{(0)}\right)^2 \phi'' \left(w_i^{(0)} x + b_i^{(0)}\right) = \left(w_i^{(0)}\right)^2 \phi'' \left(w_i^{(0)} x + b_i^{(0)}\right), \tag{20b}$$

$$\frac{\partial y''}{\partial b_i^{(0)}} = \frac{\partial}{\partial b_i^{(0)}} \sum_{i=1}^{N} w_i^{(1)} \left(w_i^{(0)}\right)^2 \phi'' \left(w_i^{(0)} x + b_i^{(0)}\right) = w_i^{(1)} \left(w_i^{(0)}\right)^2 \phi''' \left(w_i^{(0)} x + b_i^{(0)}\right), \tag{20c}$$

$$\frac{\partial y''}{\partial b^{(1)}} = 0, \tag{20d}$$

where the third derivative of the Sigmoid function (1) is required. It yields

$$\frac{d}{d\xi} \left(\phi''(\xi)\right) = \frac{d}{d\xi} \left(\phi'(\xi)(1 - 2\phi(\xi))\right) = \phi''(\xi)(1 - 2\phi(\xi)) - 2\left(\phi'(\xi)\right)^2 =$$

$$= \phi(\xi)(1 - \phi(\xi))(1 - 2\phi(\xi))^2 - 2(\phi(\xi)(1 - \phi(\xi)))^2 =$$

$$= \phi(\xi)(1 - \phi(\xi))^2 (1 - 3\phi(\xi)).$$

For $y(1)$ we obtain

$$\frac{\partial y(1)}{\partial w_i^{(0)}} = \frac{\partial}{\partial w_i^{(0)}} \left( \left( \sum_{i=1}^{N} w_i^{(1)} \phi \left(w_i^{(0)} + b_i^{(0)}\right) \right) + b^{(1)} \right) = w_i^{(1)} \phi' \left(w_i^{(0)} + b_i^{(0)}\right), \tag{22a}$$

$$\frac{\partial y(1)}{\partial w_i^{(1)}} = \frac{\partial}{\partial w_i^{(1)}} \left( \left( \sum_{i=1}^{N} w_i^{(1)} \phi \left(w_i^{(0)} + b_i^{(0)}\right) \right) + b^{(1)} \right) = \phi \left(w_i^{(0)} + b_i^{(0)}\right), \tag{22b}$$

$$\frac{\partial y(1)}{\partial b_i^{(0)}} = \frac{\partial}{\partial b_i^{(0)}} \left( \left( \sum_{i=1}^{N} w_i^{(1)} \phi \left(w_i^{(0)} + b_i^{(0)}\right) \right) + b^{(1)} \right) = w_i^{(1)} \phi' \left(w_i^{(0)} + b_i^{(0)}\right), \tag{22c}$$

$$\frac{\partial y(1)}{\partial b^{(1)}} = \frac{\partial}{\partial b^{(1)}} \left( \left( \sum_{i=1}^{N} w_i^{(1)} \phi \left(w_i^{(0)} + b_i^{(0)}\right) \right) + b^{(1)} \right) = 1, \tag{22d}$$

and we now have all the "ingredients" required to fully specify (19). To test the performance of the PINN, a Reynolds BVP was specified for a linear slider with dimensionless film thickness defined by

$$H(x) = 1 + K - Kx. \tag{23}$$

This means that $c(x) = (1 + K - Kx)^3$ and $f(x) = dH/dx = -K$ and that the exact solution is

$$y_{exact}(x) = \left[ \frac{1}{K} \left( \frac{1}{1 + K - Kx} - \frac{1 + K}{2 + K} \frac{1}{(1 + K - Kx)^2} - \frac{1}{2 + K} \right) \right], \tag{24}$$

see, e.g., [14].

The PINN (following the architecture suggested herein) was implemented in MATLAB and a numerical solution to (16) was obtained using the parameters in Table 3. As for the IVP, addressed in the previous section, the weights $w_i^{(0)}$ and bias $b_i^{(0)}$ were, again, initialised using randomly generated numbers, uniformly distributed in $[-2, 2]$, while the weights $w_i^{(1)}$ and the bias $b^{(1)}$ was initially set to zero, to ensure fulfilment of the boundary conditions.

Figure 3 depicts solution predicted by the PINN (red line with circle markers) and the exact solution obtained by integration (blue continuous line).

**Table 3.** Parameters used to defined the ANN to for the Reynolds equation.

| Parameter | Description | Value |
|-----------|-------------|-------|
| $N_i$ | # of grid points for the solution domain $[0, 1]$ | 21 |
| $K$ | Slope parameter for the Reynolds equation | 1 |
| $N_e$ | # of training batches (# or corrections during 1 epoch) | 2000 |
| $T_b$ | # of Epochs (1 epoch contains $T_b$ training batches) | 600 |
| $L_r$ | Learning rate coefficient (relaxation for the update) | 0.005 |
| $N$ | # of nodes/neurons in the hidden layer | 10 |

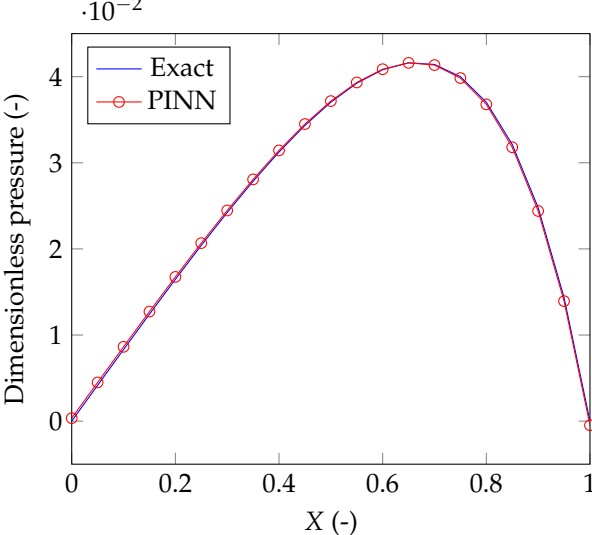

**Figure 3.** The solution achieved by the ANN (red line with circle markers) and the exact solution obtained by integration (blue continuous line).

Table 4 lists the weights an bias corresponding to the solution presented in Figure 3.

**Table 4.** Parameters used to defined the ANN.

| Node | $w^{(0)}$ | $b^{(0)}$ | $w^{(1)}$ | $b^{(1)}$ |
|------|-----------|-----------|-----------|-----------|
| 1 | 0.0557 | 1.9808 | −0.2186 | −0.0641 |
| 2 | −6.3047 | 6.1664 | 0.1220 | |
| 3 | −9.3674 | 11.4571 | 0.3843 | |
| 4 | −4.5473 | 3.3266 | 0.0305 | |
| 5 | −2.4464 | −1.9884 | 0.1188 | |
| 6 | −0.1365 | −0.1674 | 0.4155 | |
| 7 | 0.8581 | 0.5253 | 0.5089 | |
| 8 | 1.0901 | 2.0858 | 0.3348 | |
| 9 | 0.2085 | 0.2523 | −0.2024 | |
| 10 | −3.2168 | 5.9722 | −0.9899 | |

We note that, with these weights and bias, the trained network's prediction of the solution to the Reynolds BVP exhibits the overall error

$$\frac{1}{N_i} \sqrt{\sum_{k=1}^{N_i} (y_{exact}(x_k) - y(x_k))^2} = 6.2 \times 10^{-5}, \tag{25}$$

while $y(0) = 4.1 \times 10^{-4}$ and $y(1) = -4.0 \times 10^{-4}$.

**Remark 1.** *The formulation of the PINN presented here is applicable as a numerical solution procedure for the Reynolds BVP (16) and it does not consider the effect of cavitation. Exactly how the effect of cavitation can be included is, however, out of the scope of this paper.*

## 5. Concluding Remarks

A physics-informed neural network (PINN) applicable to solve initial and boundary value problems has been established. The PINN was applied to solve an initial value problem described by a first order ordinary differential equation and to solve the Reynolds boundary value problem, described by a second order ordinary differential equation. Both these problems were selected since they can be solved analytically, and the error analysis showed that the predictions returned by the PINN was in good agreement with the analytical solutions for the specifications given. The advantage of the present approach is, however, neither accuracy nor efficiency when solving these linear equations, but that it presents a meshless method and that it is not data driven. This concept may, of course, be generalised, and it is hypothesised that future research in this direction may lead to more accurate and efficient in solving related but nonlinear problems than currently available routines.

**Funding:** The author acknowledges support from VR (The Swedish Research Council): DNR 2019-04293.

**Institutional Review Board Statement:** Not applicable.

**Informed Consent Statement:** Not applicable.

**Data Availability Statement:** Not applicable.

**Conflicts of Interest:** The author declares no conflict of interest.

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
