# Peer review of "Fundamentals of Physics-Informed Neural Networks Applied to Solve the Reynolds Boundary Value Problem"

_lubricants, doi:10.3390/lubricants9080082_

Round 1

Reviewer 1 Report

Firstly thank you for submitting such an interesting paper, it was a great pleasure to read. I have never seen an AI model with this meshless approach used in such a way and with such accuracy in it's prediction, congratulations!

I have some very minor comments/questions below;

  1. The model used was very effecting at predicting the dimensionless pressure within the contact, I assume therefore it can be extended to find also the film thickness and (not as part of this paper) could it be further extended towards viscous friction calculations?
  2. The model presented was for a classical 1D solution to Reynolds equation. I wonder how the effects of different boundary conditions could be applied to the model, e.g. JFO predicting cavitation or if something like Reynolds pressure boundary condition would be applied external to the AI model.
  3. Computationally how is the performance of the AI model compared to the numerical 1D solution. I understand the comparison to analytical models, however, I would appreciate if the author can share their insight into what additional added value computationally this would have for tribodynamic simulation as I expect it to be quite large if one was to extend to a 2D Reynolds solution?

Author Response

Dear Reviewer,

Let me first thank you for reviewing my submission. I've chosen to address and answer your comments in a separate document and I hope that you will be satisfied when you take part of it.

Sincerely,
Professor Andreas Almqvist
Luleå University of Technology

Reviewer 2 Report

This paper presents a complete derivation and design of a physics-informed neural network (PINN) applicable to solve initial- and boundary value problems  described by linear ordinary di erential equations. Paper can be classified as an encouraging, proposed method is interesting. Nevertheless, there are many, in my feelings, non-completely resolved issues that make the paper complicated to clearly understand by the reader.

Author Response

Dear Reviewer, 

Let me first thank you for reviewing my submission. Since your suggestions where implemented as comments in the .pdf of the manuscript I've chosen to use the same facilities to respond. I hope that they satisfactory address and answer the questions that where raised.

Sincerely,
Professor Andreas Almqvist
Luleå University of Technology

Round 2

Reviewer 2 Report

Dear author(s), thank you for your responses. Many comments were improved significantly that makes the paper more suitable for considering to publish in the Lubricants journal.